# The Online Misinformation Susceptibility Scale: Development and Initial Validation

**DOI:** 10.3390/healthcare13172252

**Published:** 2025-09-08

**Authors:** Aglaia Katsiroumpa, Ioannis Moisoglou, Polyxeni Mangoulia, Olympia Konstantakopoulou, Parisis Gallos, Maria Tsiachri, Petros Galanis

**Affiliations:** 1Clinical Epidemiology Laboratory, Faculty of Nursing, National and Kapodistrian University of Athens, 11527 Athens, Greece; aglaiakat@nurs.uoa.gr (A.K.); pmango@nurs.uoa.gr (P.M.); olykonstant@nurs.uoa.gr (O.K.); mtsiachri@nurs.uoa.gr (M.T.); 2Faculty of Nursing, University of Thessaly, 41500 Larissa, Greece; iomoysoglou@uth.gr; 3Faculty of Nursing, University of West Attica, 12243 Athens, Greece; parisgallos@nurs.uoa.gr

**Keywords:** misinformation, susceptibility, development, validation, scale, tool, questionnaire, fake news, social media, websites

## Abstract

**Background/Objectives**: Although it is known that widespread online misinformation has negative consequences, there is no scale to measure susceptibility to online misinformation. Thus, our aim was to develop and validate a tool for measuring susceptibility to online misinformation: the Online Misinformation Susceptibility Scale (OMISS). **Methods**: A comprehensive literature review was conducted to generate a preliminary pool of items. Subsequently, a multidisciplinary panel of experts assessed the content validity of these items. To establish face validity, cognitive interviews were performed. Both exploratory and confirmatory factor analyses were performed to verify the underlying construct structure of the OMISS. We examined the concurrent validity of the OMISS by using a fake news detection scale, the Trust in Scientists Scale, a single-item trust in scientists scale, the Conspiracy Mentality Questionnaire, and a single-item conspiracy belief scale. Reliability was rigorously examined using multiple indices, including Cronbach’s alpha, McDonald’s Omega, Cohen’s kappa, and intraclass correlation coefficient. **Results**: The psychometric evaluation using both exploratory and confirmatory factor analyses supported a one-factor nine-item model for the OMISS. The OMISS demonstrated strong concurrent validity, evidenced by statistically significant correlations with the five scales mentioned above. Cronbach’s alpha and McDonald’s Omega were 0.920 and 0.921, respectively. The intraclass correlation coefficient for the OMISS was 0.994. **Conclusions**: Our comprehensive psychometric evaluations confirmed the OMISS as a valid tool for measuring online misinformation susceptibility. The OMISS holds promise as an effective tool for identifying susceptibility to misinformation and could support policymakers, health educators, healthcare professionals, and stakeholders in pinpointing high-risk groups.

## 1. Introduction

According to data from February 2025, 5.56 billion individuals worldwide are internet users, accounting for 67.9% of the global population. Furthermore, an estimated 5.24 billion people, representing 63.9% of the world’s population, are active on social media platforms [1]. Over the past 15 years, the number of internet users has steadily grown and is projected to reach 7 billion by 2029, setting a new record [2]. With over 67% of the global population now connected to the internet, the spread of false and misleading information has become increasingly effortless. In particular, on social media, the abundance of user-generated content enables people to gather around shared interests, beliefs, and narratives [3]. However, the web—particularly social media platforms—also provides fertile ground for the widespread circulation of unverified rumors [4,5,6].

Therefore, in this context, verifying information and recognizing misinformation are critical skills for improving our knowledge and making rational choices since misinformation can change perceptions and create confusion about reality [7,8]. Misinformation refers to any information that is verifiably false or misleading, irrespective of its origin or the intent behind its dissemination [9]. In other words, misinformation is false or misleading information [10]. Misinformation should be distinct from disinformation, which refers to false information that is deliberately deceptive and intentionally propagated [11,12].

While misinformation has existed for a long time, it drew significant attention in 2016 following the US Presidential Election and the UK’s Brexit referendum. During this time, completely false stories—disguised as credible news—were widely shared on social media [8]. The issue has since become even more prominent, particularly during the COVID-19 pandemic [13] and the Capitol Hill riot after the 2020 US Presidential Election [14]. Misinformation is troubling because it fosters incorrect beliefs and can deepen partisan divides, even on fundamental facts. Online, users may encounter content presented as factual that has not been verified or is simply incorrect. Just reading false news stories can make people more likely to believe them to be true later on [15]. The impact of misinformation includes a decline in the accuracy of both general knowledge and specific event details [16]. Therefore, the global spread of misinformation online has emerged as a major concern, carrying serious health, economic, political, and social implications. False and misleading content threatens both individual well-being and societal stability.

Special attention should be given to health-related online misinformation since this type of misinformation may affect rationale decision-making, health outcomes, and health behaviors such as vaccination acceptance, delaying seeking health care, and treatment adherence [17,18,19]. Recently, an umbrella review included 31 systematic reviews revealed substantial variability in the prevalence of health-related misinformation on social media, with reported estimates ranging from 0.2% to 28.8% [19]. According to this umbrella review, one of the most harmful effects of health misinformation is the rise in misinterpretation or an incorrect understanding of existing evidence. It also affects mental health, leads to inefficient use of healthcare resources, and contributes to growing vaccine hesitancy. The spread of unreliable health information can delay medical care and fuel hateful or divisive discourse. Another recent example of the direct and notable harms of misinformation to public health is the COVID-19 pandemic. Studies conducted in various countries have found a strong link between belief in COVID-19 misinformation and a lower likelihood of adhering to public health guidelines, getting vaccinated, or recommending the vaccine to others [20,21,22,23]. Experimental data have shown that exposure to vaccine-related misinformation led to a roughly 6-percentage-point drop in vaccination intent among individuals who initially stated they would “definitely accept a vaccine”, thereby threatening the achievement of herd immunity [24]. Social network analyses predict that, without action, anti-vaccine content on platforms like Facebook could dominate online discussions over the next decade [25]. Additional research has connected exposure to COVID-19 misinformation with the consumption of dangerous substances [26] and a heightened risk of violent behavior [27]. Naturally, health-related misinformation posed a public health threat well before the COVID-19 pandemic. For instance, the discredited claim linking the MMR vaccine to autism led to a notable decline in vaccination rates in the UK [28].

So far, the only method available to assess individuals’ vulnerability to online misinformation is through news evaluation tasks [29,30,31,32,33,34]. In these tasks, participants are shown social media posts or news headlines—both real and fake—and are asked to judge their authenticity. Researchers typically either generate these items using common misinformation tactics [35,36] or select them from trusted fact-checking databases [13,29,37,38,39,40]. The participants then evaluate the reliability or accuracy of the content using a Likert scale or a binary scale (e.g., true vs. false) [41]. In some cases, the news is displayed as plain text [20], while in others, it is accompanied by elements such as an image, a source, and a lede sentence [29]. The ratio of true to false items varies across studies; some include only false news [20], while others present an unbalanced [42] or balanced [29] mix. Researchers often compute an index score by averaging all the ratings (to reflect the general belief in true or false information) [35]. Also, scholars may calculate the difference between the ratings of true and false items, which indicates a person’s ability to distinguish between accurate and inaccurate news (veracity discernment) [13]. Moreover, in misinformation research, it is common for scholars to assume—rather than know—that they are measuring the same underlying construct [33]. If this assumption proves incorrect, there is a risk of misrepresenting different psychological processes as a single mechanism, leading to an illusory essence bias [43]. Given the complexity of misinformation, responses to one set of items may stem from motivational influences, while responses to another may reflect critical thinking ability, suggesting these tools may not all be assessing the same “discernment skill” [33]. Currently, we lack a clear understanding of how various online misinformation susceptibility measures relate to each other, how different true-to-false ratios affect results [44], and to what extent the observed effects stem from response biases rather than genuine skill differences [45]. The limited literature that has investigated scale-specific effects has revealed significant item-related influences, raising concerns about the validity of conclusions regarding the effectiveness of interventions [42]. Additionally, issues such as the ecological validity of the item selection, presentation format, and response methods are often overlooked [41,46]. As a result, it is challenging to determine whether differences between studies arise from variations in conceptual frameworks, measurement scales, or actual differences in susceptibility to misinformation. The substantial methodological diversity across studies hinders meaningful comparison and limits the generalizability of findings, highlighting the urgent need for a reliable and valid measurement instrument.

Moreover, news evaluation tasks directly assess a participant’s knowledge of fake news, but this focus also presents a drawback. These tasks rely on particular social media posts or online news headlines, which are inherently tied to their unique historical and cultural contexts. Consequently, the generalizability of findings may be limited, as performance in such tasks could vary across different temporal or socio-cultural settings. For instance, Maertens et al. [33] performed their study during 2022 with a sample of US residents and several items are irrelevant to non-US citizens, who therefore require a different test. Additionally, several items may not be appropriate even for US residents after a few years since new data and information may emerge (e.g., “New study: clear relationship between eye color and intelligence”). Similarly, Arin et al. [32] collected their data in 2020 from citizens in Germany, the United Kingdom, France, and Spain, but several items on their scale were not up to date. Therefore, since each study uses different posts from various time periods, the built-in context of these items confines their relevance to particular locations, cultures, or timeframes making it difficult to compare results across studies.

Online misinformation endangers science, democracy, and rational behavior. Moreover, in clinical and community settings, misinformation is a great threat to public health, healthcare services, and patient outcomes. In this context, we developed a new tool for assessing public susceptibility to online misinformation: the Online Misinformation Susceptibility Scale (OMISS). To the best of our knowledge, the OMISS is the first instrument that measures individuals’ behaviors toward online information verification. Although extensive research [29,30,31,32,33,34] has been conducted, no valid instrument currently exists to assess people’s behaviors toward online misinformation. In particular, until now, all other scales have assessed individuals’ knowledge by measuring their ability to identify fake and true online news. We extend the current literature and improve our ability to verify valid online information and recognize misinformation by developing the OMISS. Unlike other similar instruments, the OMISS is a self-report instrument designed for general application across diverse web user populations. The OMISS measures general behaviors toward online misinformation and not specific knowledge like the other scales used in this field. Moreover, the OMISS is a general-context measure that can measure online misinformation susceptibility in all fields. Therefore, the OMISS is a new valid tool that we can use to fight online misinformation and fake news in every domain of our lives such as healthcare, science, politics, and news media.

In brief, the aim of this study was to develop and validate a tool to measure public susceptibility to online misinformation. Our hypothesis was that the Online Misinformation Susceptibility Scale is a reliable and valid tool that can be used by policymakers, health educators, healthcare professionals, and stakeholders to accurately measure people’s susceptibility to online misinformation.

## 2. Materials and Methods

### 2.1. Development of the Scale

We applied the suggested guidelines [47,48,49] to develop and validate the Online Misinformation Susceptibility Scale. Figure 1 shows the development and validation of the OMISS.

A systematic review was conducted to generate a preliminary pool of items. We searched for articles on online misinformation, fake news, and disinformation [4,29,30,31,32,33,50,51,52]. In particular, we searched on Pubmed and Scopus using the following strategy for titles: (((misinformation) OR (“fake news”)) OR (disinformation)) OR (malinformation). We employed the Preferred Reporting Items for Systematic Reviews and Meta-Analysis (PRISMA) guidelines to conduct our review [53]. We examined studies that explore the issues of misinformation, fake news, disinformation, and malinformation. Moreover, we examined articles published in English and in journals with a peer review system. We searched Pubmed and Scopus from their inception until 10 May 2025. We identified 2686 articles through Pubmed and 13,620 articles through Scopus. Emphasis was given to articles that measure misinformation, fake news, disinformation, or malinformation. Three independent authors performed the review. First, they removed duplicates and then examined the titles and abstracts to identify relevant articles. Our inclusion criteria were as follows: (a) articles that examined misinformation, fake news, disinformation, or malinformation; (b) articles in English; and (c) articles in journals with a peer review system. Three independent authors extracted data from studies, such as the authors, year of publication, country, study design, study population, sampling method, data collection time, and results. The Joanna Briggs Institute critical appraisal tools [54] were employed to examine the quality of the studies. These risk of bias tools examine several aspects of studies quality, such as the study setting, inclusion criteria, methods for measuring the exposure and outcome, measurement of confounding variables, and statistical analysis. Then, the quality of the studies was classified as poor, moderate, or high. Our aim was to create a preliminary list of items for our scale. Finally, we selected 26 items to develop our tool for measuring susceptibility to online misinformation (Appendix A).

Then, a panel of 10 experts with diverse professional backgrounds—including social scientists, psychologists, journalists, and experts on political science, computer science, communications, and media studies—was assembled to assess the content validity of the initial set of 26 items. Each expert was instructed to evaluate the extent to which every item aligned with the concept of online misinformation susceptibility using a three-point rating scale: not essential, useful but not essential, or essential. We calculated the content validity ratio (CVR) for each item using the following formula:$$Content\;validity\;ratio=\frac{n-\frac{N}{2}}{\frac{N}{2}}$$

In the formula above, *n* denotes the number of experts who rated an item as “essential,” while *N* refers to the total number of experts participating in the evaluation, which in this study was fixed at 10. Following the recommendations in the literature, only the items exhibiting a CVR exceeding 0.80 [55] were retained. Following this evaluation, 7 items were eliminated, resulting in a refined scale comprising 19 items.

Subsequently, the face validity of the OMISS was assessed through cognitive interviews [56] with five participants. All the participants interpreted the 19 items as intended, confirming their conceptual alignment with the study’s objectives. To further evaluate item clarity, a pilot study was conducted with 15 participants (7 males and 8 females; mean age: 36.7 years) who rated each item on a four-point Likert scale: 1 = item is not clear; 2 = item is somewhat clear; 3 = item is quite clear; and 4 = item is highly clear. Then, we calculated the item-level face validity index, retaining only the items exceeding the recommended threshold of 0.80 in accordance with established guidelines [57]. The item-level face validity index ranged from 0.933 to 1.000, indicating high clarity across all items. Consequently, all 19 items were retained in the scale after the examination of the content and face validity of the OMISS (Appendix A).

### 2.2. Research Design

We conducted a cross-sectional study. Our sample included adults aged 18 years or older. Additionally, the eligible participants were required to engage with social media platforms and/or browse websites for a minimum of 30 min per day. An anonymous online questionnaire was developed using Google Forms and disseminated through multiple digital platforms, including Facebook, Instagram, and LinkedIn. To further broaden participation, a promotional video was created and shared on TikTok, inviting users to take part in the study. Data collection was conducted from 24 June 2025 until 1 July 2025.

Our final sample included 522 participants. Among them, 79.7% (n = 416) were female and 20.3% (n = 106) were male. The mean age of the participants was 37.8 years (standard deviation = 13.0), with a median of 38 years and an age range spanning from 18 to 80 years. The mean daily usage time for social media/websites was 3.3 h (standard deviation: 2.1), with a median value of 3 h and a range spanning from 30 min to 10 h.

### 2.3. Item Analysis

Following the initial development phase of the OMISS, an item analysis was conducted on the 19 generated items. The analysis was performed using the full sample (n = 522) to assess inter-item correlations, corrected item-total correlations, floor and ceiling effects, skewness, kurtosis, and Cronbach’s alpha (when a single item was deleted) for our 19 items [58]. Inter-item correlations were evaluated against the recommended threshold of 0.15 to 0.75 [59]. Corrected item-total correlations were required to exceed 0.30 to ensure adequate discriminatory power [60].

We asked the participants to think about what they do when they see a post or story that interests them on social media or websites. The items included the following: (1) How often do you check the website domain and URL? (2) How often do you check the publication date of the post? (3) How often do you check if the post includes reliable links and references such as scientific articles? (4) How often do you check if the post includes the author’s name?

The participants rated each item on a five-point Likert scale: never, rarely, sometimes, very often, and always. Floor and ceiling effects were considered to be present if more than 85% of the respondents selected the lowest (“never”) or highest (“always”) possible score on the five-point Likert scale [61]. The normality of the distribution was assessed using skewness (acceptable range: −2 to +2) and kurtosis (acceptable range: −3 to +3) [62].

### 2.4. Construct Validity

To evaluate the construct validity of the OMISS, both exploratory factor analysis (EFA) and confirmatory factor analysis (CFA) were conducted. The literature recommends a minimum sample size of 50 observations or five observations per item for EFA [63], and at least 200 observations for CFA [58]. Our sample satisfied these requirements. Specifically, the participants were randomly divided into two independent subsamples so that the EFA and CFA could be conducted on separate datasets. Specifically, we randomly divided our participants into two distinct groups: one sample of 261 participants for the EFA and another sample of 261 participants for the CFA. This strategy of employing separate samples for each analysis was implemented to enhance the robustness and validity of our psychometric evaluations. Both subsamples exceeded the established minimum sample size thresholds for our analyses.

Initially, EFA was conducted to identify the underlying factor structure of the OMISS, followed by CFA to validate the structure derived from the EFA. This stage included the 12 items retained after the initial development and item analysis of the OMISS.

Prior to conducting the EFA, we evaluated the suitability of our data using the Kaiser–Meyer–Olkin index and Bartlett’s test of sphericity. Following conventional psychometric standards, we established the following acceptability thresholds: Kaiser–Meyer–Olkin index exceeding 0.80 and Bartlett’s test achieving statistical significance (*p* < 0.05) [62]. For the EFA, we implemented oblique rotation (Promax method in IBM SPSS 28.0). The following are the acceptable values for the EFA: eigenvalues > 1, factor loadings > 0.60, communalities > 0.40, and >65% of the total variance can be explained by the factors [62]. Furthermore, we assessed the internal consistency of the factors by computing Cronbach’s alpha coefficients, with values above 0.70 considered indicative of acceptable reliability [64].

Then, we validated the factor structure of the OMISS through CFA. Given that the OMISS responses showed a normal distribution, the maximum likelihood estimator was employed for the analysis. To assess the model fit in the CFA, we evaluated multiple goodness-of-fit indices, including two absolute fit measures—the Root Mean Square Error of Approximation (RMSEA) and the Goodness of Fit Index (GFI)—along with two relative fit indices: the Normed Fit Index (NFI) and the Comparative Fit Index (CFI). Additionally, one parsimonious fit measure—the ratio of chi-square to the degrees of freedom (x^2^/df)—was examined. The established thresholds for an acceptable model fit are an RMSEA < 0.10, GFI > 0.90, NFI > 0.90, CFI > 0.90, and x^2^/df < 5 [65,66]. Furthermore, we computed the standardized regression weights between the items and factors.

Finally, we examined the measurement invariance across demographic groups. In particular, we examined the measurement invariance between groups based on gender, age, and daily usage time for social media/websites. We used median values to divide the participants into two groups according to their age and daily usage time for social media/websites. We tested configural invariance and then metric invariance [67]. In the case of configural invariance, we examined the RMSEA, GFI, NFI and CFI values. The acceptable values were as follows: less than 0.10 for RMSEA, and more than 0.90 for GFI, NFI, and CFI. In the case of metric invariance, a *p*-value higher than 0.05 indicated metric invariance.

### 2.5. Concurrent Validity

After examining the construct validity, we identified a one-factor 9-item model for the OMISS. The participants rated each item on a five-point Likert scale ranging from never to always. We assigned a score for each response, with higher scores indicating greater misinformation susceptibility. Thus, we assigned a value of 1 to “always”, 2 to “very often”, 3 to “sometimes”, 4 to “rarely”, and 5 to “never”. The total score for the OMISS ranges from 9 to 45. Higher scores indicate higher misinformation susceptibility.

We measured the concurrent validity of the OMISS through several methods. Since there are no other scales that measure misinformation susceptibility, we employed several other scales to measure the concurrent validity of the OMISS.

First, considering that higher levels of online misinformation susceptibility could be correlated with a lower ability to detect fake news [20,29,41], we measured the participants’ ability to detect fake news. Thus, we provided the participants with two fake news headlines and two true news headlines. Since we conducted this study in Greece, we selected news headlines that avoided any national bias for this country. Thus, we excluded headlines referring to domestic affairs in Greece and national policy stances. Moreover, we presented the headlines in text form. We chose news headlines that were up to date at the time the study was conducted. We pre-tested the four headlines among 15 individuals to confirm that we received varying responses. The two fake news headlines were taken from the DiSiNFO Database [68]. The DiSiNFO Database is part of the Diplomatic Service of the European Union [69], which consists of experts in journalism, communications, and social sciences. The objective of the DiSiNFO Database is to improve individuals’ awareness and understanding of misinformation and fake news. The DiSiNFO Database uses media-monitoring services and data analysis in 15 languages to identify, compile, and expose misinformation cases. The selected fake headlines were (1) “US intelligence services read the messages of WhatsApp users” and (2) “Western mass media and social media are controlled by the government”. Additionally, one true news headline was taken from *The Guardian* [70] and one from *France 24* [71]. *The Guardian* is a British daily newspaper that is available online. *The Guardian* is a “newspaper of record”, i.e., a newspaper with a prominent national publication with a wide readership due to its authoritative and impartial journalism. “Newspapers of record” earn their status through reputation and are often among the oldest and most esteemed in the world [72]. *France 24* is a Paris-based international news network funded by the French government. It broadcasts in French, English, Arabic, and Spanish, primarily targeting global audiences. The channel is accessible worldwide via satellite and offers live news streaming through its website, YouTube, and mobile/digital platforms [73]. The selected true news headlines were (1) “NASA data reveals dramatic rise in intensity of weather events” and (2) “French President Emmanuel Macron wants to ban social media for under-15 s in France”. We asked the participants, “To the best of your knowledge, how likely is it that the claim in each headline is correct?” The answers were given using a five-point Likert scale: (a) extremely unlikely, (b) somewhat unlikely, (c) neither unlikely nor likely, (d) somewhat likely, and (e) extremely likely. For the two fake news headlines, we assigned a value of 4 for the answer “extremely unlikely”, 3 for “somewhat unlikely”, 2 for “neither unlikely nor likely”, 1 for “somewhat likely”, and 0 for “extremely likely”. We used the reverse coding for the two true news headlines. Then, we summed the answers on the four news headlines and calculated a total score ranging from 0 to 16. Higher scores indicated better fake news detection. We expected a negative correlation between scores for fake news detection and scores on the OMISS.

Second, considering that low trust in scientists is related to negative behaviors such as vaccine hesitancy [24,74], views on climate change [75,76], and exposure to untrustworthy websites [77], we measured the participants’ trust in scientists. We used the Trust in Scientists Scale [78] to measure the participants’ trust in scientists. The Trust in Scientists Scale comprises 12 items and measures four dimensions of trust in scientists: integrity, competence, benevolence, and openness. Since these four factors are highly correlated, we chose one of them (integrity) to use in this study. Integrity was measured using three items (e.g., How honest or dishonest are most scientists?), and the answers were given on a five-point Likert scale ranging from 1 (very dishonest) to 5 (very honest). The total score is calculated as the mean of the answers on the three items. Thus, the total integrity score ranges from 1 to 5. Higher values indicate higher levels of trust in scientists. We used the valid Greek version of the Trust in Scientists Scale [78]. In this study, the Cronbach’s alpha for the integrity scale was 0.897 and the McDonald’s Omega was 0.904. We expected a negative correlation between scores on the integrity scale and scores on the OMISS.

Third, we used a single question taken from the Pew Research Center to further measure the participants’ level of confidence in scientists [79]. This question was “How much confidence do you have in the scientists to act in the best interests of the public?” The answers were given on a five-point Likert scale ranging from 1 (no confidence at all) to 5 (a great deal of confidence). Higher scores indicate higher levels of trust in sciences. The Pew Research Center is an impartial research organization dedicated to providing objective information about global trends, public attitudes, and societal issues. Through comprehensive public opinion surveys and demographic studies, the Pew Research Center delivers data-driven insights to inform public discourse. Maintaining strict neutrality, the organization refrains from advocating for any policy positions or political stances [80]. We expected a negative correlation between trust in scientists and scores on the OMISS.

Fourth, the literature suggests that conspiracy theories erode public trust in scientifically validated positions, particularly regarding vaccination programs [81,82], climate change [83,84], and the COVID-19 pandemic [81,85]. Thus, we measured the generic conspiracy beliefs of our participants using the Conspiracy Mentality Questionnaire (CMQ) [86]. The CMQ comprises five items, and responses are given on an 11-point Likert scale ranging from 0 (totally disagree) to 10 (totally agree). The total score on the CMQ is calculated by adding answers to the five items. Therefore, the total score on the CMQ ranges from 0 to 50, with higher values indicating stronger conspiracy beliefs. We used the valid Greek version of the CMQ [87]. In this study, the Cronbach’s alpha for the CMQ was 0.865 and the McDonald’s Omega was 0.880. We expected a positive correlation between scores on the CMQ and scores on the OMISS.

Fifth, we used a single-item conspiracy belief scale; the item is “I think that the official version of the events given by the authorities very often hides the truth” [88]. The answers are given using a nine-point Likert scale ranging from 1 (completely false) to 9 (completely true). Higher scores indicate a stronger belief in conspiracy theories. We expected a positive correlation between scores on the single-item conspiracy belief scale and scores on the OMISS.

### 2.6. Reliability

We calculated the Cronbach’s alpha and McDonald’s Omega for the OMISS to evaluate the internal consistency of the scale. We used the overall sample (n = 522) to calculate these indexes. Cronbach’s alpha and McDonald’s Omega values higher than 0.7 are considered acceptable values [64].

Moreover, we calculated corrected item-total correlations and Cronbach’s alpha when a single item was deleted from the complete OMISS (with nine items). Corrected item-total correlations higher than 0.30 were considered acceptable [60].

Finally, we conducted a test–retest study with 40 participants that answered the OMISS twice in five days. We calculated the Cohen’s kappa for the nine items of the OMISS since the answers were given on a five-point ordinal scale. Additionally, we measured the two-way mixed intraclass correlation coefficient (absolute agreement) for the total OMISS scores.

### 2.7. Ethical Considerations

The study employed an anonymous, voluntary data collection procedure. Prior to participation, all individuals received comprehensive information regarding the research objectives and methodology, after which, they provided informed consent. Ethical approval for this study was obtained from the Institutional Review Board at the Faculty of Nursing, National and Kapodistrian University of Athens (Protocol Approval #55, 23 June 2025). The study was conducted in full compliance with the ethical principles outlined in the Declaration of Helsinki [89].

### 2.8. Statistical Analysis

Categorical variables were summarized using absolute frequencies (n) and the corresponding percentages. For continuous variables, measures of central tendency (mean and median) and dispersion (standard deviation and range) were reported, along with minimum and maximum values. The normality of the scale distributions was assessed using both statistical methods (Kolmogorov–Smirnov test) and graphical techniques (Q–Q plots). Thus, correlations between scales were examined using Pearson’s correlation coefficients. Moreover, we calculated the effect sizes between scales using Pearson’s correlation coefficients. In particular, we squared the correlation coefficient values to calculate the effect size. We transformed these values into percentages for easier comprehension. The squared correlation is the coefficient of determination, which indicates the proportion of variance in one variable that is explained by variance in another variable. *p*-values less than 0.05 were considered statistically significant. Before entering the data into IBM SPSS 28.0, we carefully examined the participants’ answers on Google Forms to identify possible inconsistencies. We performed CFA using AMOS version 21 (Amos Development Corporation, 2018, Armonk, NY, USA). All other analyses were conducted using IBM SPSS Statistics for Windows, Version 28.0 (released 2012; IBM Corp., Armonk, NY, USA).

## 3. Results

### 3.1. Item Analysis

Table 1 presents the results from the item analysis. We deleted items #2, #3, #6, #11, #12, #18, and #19 due to unacceptable values in several criteria in the item analysis. In particular, these items had corrected item-total correlations lower than 0.30 and showed floor effects (Table 1). Moreover, these items showed negative inter-item correlations or inter-item correlations outside of the acceptable range of 0.15 to 0.75 with several other items (Appendix A). Items #2, #3, #6, #11, #12, #18, and #19 did not show a normal distribution, with skewness and kurtosis values outside of the range of −2 to +2 and −3 to +3, respectively (Table 1).

The Cronbach’s alpha for the 19 items was 0.888. Removal of item #2, #3, #6, #11, #12, #18, or #19 increased the Cronbach’s alpha.

Thus, we deleted seven items (#2, #3, #6, #11, #12, #18, and #19; Table 1), and the remaining 12 items had acceptable corrected item-total correlations, inter-item correlations, floor and ceiling effects, skewness, and kurtosis. The Cronbach’s alpha for the 12 items was 0.920. Appendix A shows the 12 items that were selected after the item analysis for the OMISS.

### 3.2. Exploratory Factor Analysis

The Kaiser–Meyer–Olkin measure of the sampling adequacy was 0.891, and the Bartlett’s test of sphericity result was statistically significant (*p* < 0.001), confirming the suitability of the data for conducting EFA. An oblique rotation using the Promax method was used in the EFA, which included the 12 items described above (items #1, #4, #5, #7, #8, #9, #10, #13, #14, #15, #16, and #17; Table 1).

Table 2 shows the EFA results. Three items (#1, #8, and #12) had unacceptable factor loadings and communalities. In particular, these three items had factor loadings lower than 0.60 and communalities lower than 0.40. The EFA identified one factor that explained 50.46% of the total variance. Thus, we conducted EFA again after removing items #1, #8, and #12 and compared the findings of the two models. The Kaiser–Meyer–Olkin measure of sampling adequacy was 0.926 and the Bartlett’s test of sphericity result was statistically significant (*p* < 0.001), confirming the suitability of the data for conducting EFA. Table 3 shows the results from the second EFA. Now, all the items had factor loadings higher than 0.723 and communalities higher than 0.523. Moreover, the EFA identified one factor that explained 64.47% of the total variance. Thus, the second model shown in Table 3 was better than the first model shown in Table 2.

The OMISS demonstrated excellent internal consistency, with a Cronbach’s alpha of 0.920 and McDonald’s Omega of 0.931.

Appendix A shows the nine items that were selected after the exploratory factor analysis for the OMISS.

### 3.3. Confirmatory Factor Analysis

CFA was conducted to validate the one-factor structure of the OMISS developed using EFA. The CFA included nine items loaded into a single factor. The model fit indices indicated an excellent fit to the data (x^2^/df = 1.437, RMSEA = 0.041, GFI = 0.976, NFI = 0.979, CFI = 0.993). The standardized factor loadings for the nine items ranged from 0.63 to 0.84, which were all statistically significant (*p* < 0.001). Figure 2 shows the CFA results for the OMISS.

In conclusion, our factor analysis identified a one-factor nine-item model for the OMISS (Appendix A).

### 3.4. Concurrent Validity

We found a negative correlation between the fake news detection scale and the OMISS scores, suggesting that participants with a lower ability to detect fake news have higher levels of misinformation susceptibility (r = −0.135, *p*-value = 0.002, effect size = 1.83%).

Furthermore, a significant negative correlation was observed between the Trust in Scientists Scale and OMISS scores (r = –0.304, *p* < 0.001, effect size = 9.24%), suggesting that a lower trust in scientists is associated with higher susceptibility to misinformation. A similar negative correlation was found between the single-item trust in scientists measure and OMISS scores (r = −0.280, *p*-value < 0.001, effect size = 7.84%) (Table 4).

Moreover, we found a positive correlation between the Conspiracy Mentality Questionnaire and OMISS scores, suggesting that participants with stronger conspiracy beliefs are also have higher susceptibility to misinformation (r = 0.159, *p*-value < 0.001, effect size = 2.53%). Similarly, the single-item conspiracy belief scale scores were positively associated with the OMISS scores (r = 0.095, *p*-value = 0.030, effect size = 0.91%) (Table 4).

Thus, the concurrent validity of the OMISS was excellent.

### 3.5. Measurement Invariance

Table 5 shows the configural measurement invariance for the one-factor CFA model with respect to gender, age, and daily usage time for social media/websites. The RMSEA, GFI, NFI, and CFI values for these three demographic characteristics indicated that the one-factor CFA model was a good fit for all groups, such as females vs. males, younger vs. older participants, and participants with lower daily usage time for social media/websites vs. those with higher daily usage time for social media/websites. Additionally, our findings supported metric invariance since the *p*-values for gender, age, and daily usage time for social media/websites were 0.454, 0.068, and 0.303.

### 3.6. Reliability

For the one-factor model comprising nine items, the Cronbach’s alpha and McDonald’s Omega were 0.920 and 0.921, respectively, indicating excellent internal consistency. The corrected item-total correlations ranged from 0.625 to 0.815, and the removal of any single item did not improve the Cronbach’s alpha (Appendix A). The intraclass correlation coefficient (ICC) for the OMISS was 0.994 (95% CI: 0.989–0.997; *p* < 0.001). Additionally, the Cohen’s kappa values for the nine items ranged from 0.732 to 0.968 (*p* < 0.001 for all), as shown in Appendix A. These findings collectively demonstrate the excellent reliability of the OMISS.

## 4. Discussion

In this study, we developed and validated the Online Misinformation Susceptibility Scale to assess online misinformation susceptibility. Our thorough validity and reliability analyses suggest that the OMISS is a reliable and valid tool for measuring susceptibility to misinformation. Moreover, the OMISS includes nine items and only requires a few minutes to complete. Therefore, the OMISS is a brief and easy-to-use tool with robust psychometric properties.

Until now, there is only one approach to measure individuals’ vulnerability to online misinformation, namely “news evaluation tasks” [29,30,31,32,33,34]. Briefly, by applying “news evaluation tasks”, researchers create instruments that evaluate individuals’ ability to detect true and fake social media posts or online news headlines. The participants evaluate the accuracy of social media posts or online news headlines through a Likert scale or a binary scale (e.g., true vs. false) [41]. Then, researchers compute a total knowledge score by averaging all the ratings for the social media posts or news headlines that the instrument includes [13,20,35,42]. However, this approach has several limitations: (1) Misinformation is a complex issue and answers to one instrument may stem from motivational influences, while responses to another instrument may reflect critical thinking ability; thus, these instruments may not all be assessing the same “discernment skill” [33]. (2) The scales have low ecological validity due to the item selection, presentation format, and response methods [41,46]. (3) The scales based on “news evaluation tasks” include social media posts or online news headlines, which are inherently tied to their unique historical and cultural contexts [32,33]. (4) The substantial methodological diversity across studies hinders meaningful comparisons and limits the generalizability of the findings [29,30,31,32,33,34].

To overcome these limitations, we developed the OMISS. To the best of our knowledge, the OMISS is the first self-report instrument that measures individuals’ behaviors toward online information verification. We believe that the OMISS could be used to improve our ability to verify valid online information and recognize misinformation. Unlike other similar scales, the OMISS was designed for general application across diverse web user populations. Our scale measures general behaviors toward online misinformation and not specific knowledge like other scales [29,30,31,32,33,34].

Moreover, the OMISS has another important property. As we discussed before, the OMISS is a general-context measure that can measure online misinformation susceptibility in all fields. Additionally, with very slight adaptation, scholars can use the OMISS to measure online misinformation in specific domains. In particular, the introductory note in the OMISS states the following: “Please think about what you do when you see a post or story that interests you on social media or websites”. In this way, we can measure misinformation susceptibility in all fields. In a study where researchers want to measure health-related misinformation susceptibility, they can adjust the introductory note by adding two simple words, i.e., “Please think about what you do when you see a health-related post or story that interests you on social media or websites”. After this slight adjustment, the OMISS now only refers to health-related online misinformation and not to online misinformation in general. Similarly, if the aim of the study is to assess science-related misinformation susceptibility, then the introductory note in the OMISS should be “Please think about what you do when you see a science-related post or story that interests you on social media or websites”. Another common type of misinformation is political misinformation. In this case, scholars should adjust the OMISS to “Please think about what you do when you see a political-related post or story that interests you on social media or websites”.

The measurement of misinformation susceptibility using valid scales such as the OMISS could offer several important scientific, social, and practical benefits. It could help individuals, organizations, and governments respond more effectively to one of the most pressing challenges in the information age. In particular, the OMISS could help us to improve our ability to identify who is more likely to believe misinformation. In other words, we can measure who is most susceptible to misinformation (e.g., based on age, education, political identity, or social media use). Then, policymakers can develop and apply evidence-based campaigns to reduce online misinformation (e.g., education on vaccine safety and effectiveness, election integrity, and health behaviors) by creating interventions to target specific personality traits, thinking styles, or emotional triggers [51,90]. For instance, the World Health Organization uses susceptibility metrics to prioritize health misinformation responses through its infodemic management campaign [91]. Also, governments, health agencies, and media can adjust messaging strategies to minimize misunderstandings. For instance, targeted media literacy programs from authoritative organizations such as the European Commission have proven to be effective in reducing health-related misinformation on several issues such as the COVID-19 pandemic and vaccine hesitancy [92]. Moreover, researchers could use the OMISS to assess whether implemented interventions and educational programs actually reduce the belief in false information. Furthermore, the OMISS could enable interdisciplinary research in healthcare sciences, communications, political science, and public health since researchers could use it to measure misinformation susceptibility in these fields.

We followed the suggested guidelines [47,48,49] to develop and validate the OMISS. In brief, (a) we examined the content validity by calculating the content validity ratio; (b) we assessed the face validity by performing cognitive interviews and calculating the item-level face validity index; (c) we performed item analysis; (d) we examined the construct validity by performing EFA and CFA; (e) we assessed the concurrent validity by using five other valid scales; and (f) we evaluated reliability through Cronbach’s alpha, McDonald’s Omega, Cohen’s kappa, and intraclass correlation coefficients.

This study had several limitations. First, we used five scales to examine the concurrent validity of the OMISS: (1) a fake news detection scale, (2) the Trust in Scientists Scale, (3) a single-item trust in scientists scale, (4) the Conspiracy Mentality Questionnaire, and (5) a single-item conspiracy belief scale. The concurrent validity of the OMISS can be further explored by using measures of cognitive effort (e.g., analytical reasoning [93]) since there is a link between analytical reasoning and critical thinking [30]. Moreover, scholars could use motivation to engage participants in cognitive activities since the literature suggests an association between engagement in cognitive activities and scientific interest [94], partisan evaluations [95], and innovation [96]. Second, the OMISS, as well as the other scales we used in this study, were self-report instruments; thus, information bias was likely present due to some participants’ subjectivity. Our participants may provide answers they believe are socially acceptable rather than their true thoughts or behaviors, leading to biased results. Moreover, self-reports capture perceptions rather than objective facts, which may not accurately reflect actual behaviors or conditions that affect their susceptibility to misinformation. Future studies should employ longitudinal validation or the use of digital behavior data to verify the self-reports. Third, this study was carried out in a specific country using a convenience sample. For example, the percentage of male participants was relatively low compared to the general population. As a result, the findings cannot be broadly generalized, and future research should aim to use more representative samples and diverse populations to further evaluate the validity of the OMISS. Nevertheless, the psychometric analysis remains robust, as the sample size fulfilled all the necessary criteria. In any case, scholars should examine the psychometric properties of the OMISS in different populations and cultural settings. Fourth, we used a cross-sectional study to assess the validity of the OMISS. However, because the participants’ behaviors may vary over time, longitudinal research is needed to explore how susceptibility to misinformation evolves. Fifth, we performed a thorough examination of the psychometric properties of the OMISS, but other scholars could use other approaches to examine the reliability and validity of the scale, such as divergent validity, criterion validity, and known-groups validity. Sixth, as this was the first assessment of the validity of the OMISS, we did not aim to determine a cut-off score. Future research could conduct cut-off analyses to identify threshold values for distinguishing between participants. Seventh, we employed a convenience sample that was collected through social media to develop and validate the OMISS. Recruitment through social media can introduce sampling bias because the individuals reached via these platforms are not necessarily representative of the broader target population. For instance, people who respond are often more digitally active or interested in the study topic, thereby excluding less engaged individuals. Also, platform-specific demographics can skew the sample since different social media platforms attract users with distinct age, socioeconomic, and cultural profiles. Moreover, the exclusion of non-users systematically omits individuals without social media access, limiting the generalizability. Finally, we developed and validated the OMISS using a sample from one country. Expanding the utility of the OMISS across different cultural and linguistic contexts requires further studies to ensure the reliability, validity, and cultural sensitivity of the scale. The OMISS should be culturally adapted and validated in populations with diverse sociocultural backgrounds to broaden its scope and generalizability. Scholars should also slightly modify some items to reflect culturally relevant expressions, norms, and practices without altering the underlying construct. For example, scholars should conduct cognitive interviews with participants from the target culture to ensure clarity and appropriateness.

## 5. Conclusions

Online users are increasingly exposed to misinformation at an accelerating pace. To separate fact from falsehood, today’s audiences need strong information literacy skills that enable them to verify the content they encounter. Although the importance of verifying online information is widely recognized, there are currently no established tools to assess this ability. Additionally, researchers lack a consistent definition of misinformation susceptibility and often rely on invalid measurement tools. To the best of our knowledge, there is no scale to measure susceptibility to online misinformation. To address this gap, this study developed and validated the Online Misinformation Susceptibility Scale. The OMISS is the first valid tool that enables scholars, policymakers, and stakeholders to accurately measure levels of online misinformation. We believe our findings offer valuable insights into misinformation susceptibility and provide relevant guidance for policymakers aiming to limit the spread of misinformation and lower the public’s vulnerability to it.

Following a comprehensive analysis of its reliability and validity, we found that the OMISS is a brief, user-friendly tool with robust psychometric properties. Our findings suggest that the one-factor nine-item OMISS that can measure public misinformation susceptibility. As such, it serves as a timely instrument for measuring susceptibility levels and identifying individuals at higher risk. Given the study’s limitations, we recommend translating the OMISS into and validating it in different languages and populations to further assess its reliability and validity. The OMISS holds promise as an effective tool for identifying susceptibility to misinformation and could support policymakers, health educators, healthcare professionals, and stakeholders in pinpointing high-risk groups. For example, researchers and practitioners can use the OMISS to evaluate the effectiveness of media literacy interventions by comparing the scores before and after the interventions. Furthermore, researchers in all disciplines (e.g., healthcare scientists, social scientists, political scientists, and journalists) can use the Online Misinformation Susceptibility Scale to identify predictors and consequences of misinformation susceptibility.

Identification of the level of online misinformation susceptibility can help people in several ways. For instance, the incorporation of the OMISS into digital literacy interventions could offer a structured and evidence-based approach to enhancing program effectiveness. Initially, the OMISS can be administered as a baseline assessment to determine the participants’ susceptibility to misinformation, thereby enabling the identification of knowledge gaps and cognitive vulnerabilities. This diagnostic function could facilitate the customization of instructional content, ensuring that the intervention modules address the specific needs of specific participant groups. For instance, individuals exhibiting higher susceptibility could receive targeted training on fact-checking techniques, source verification, and critical evaluation of online content. During the intervention, selected items from the OMISS can be embedded into interactive learning activities, such as quizzes and scenario-based exercises, to provide real-time feedback and reinforce critical thinking skills. This dynamic integration not only enhances learner engagement but also promotes the practical application of misinformation detection strategies in simulated digital environments. Following the intervention, the OMISS could be re-administered to evaluate the program’s impact, allowing for the measurement of knowledge gains and behavioral changes.

Furthermore, the OMISS can serve as a tool for continuous monitoring by incorporating periodic assessments within digital platforms or learning management systems. This longitudinal approach enables the tracking of sustained improvements in misinformation resilience and informs iterative refinements to program content. On a broader level, aggregated and anonymized data derived from OMISS administration can provide valuable insights into demographic or regional patterns of misinformation susceptibility, thereby informing policy development and large-scale digital literacy strategies. By embedding the OMISS throughout the intervention cycle, practitioners can ensure that digital literacy programs are both adaptive and empirically grounded, ultimately contributing to more resilient and informed digital citizens.

## Figures and Tables

**Figure 1 healthcare-13-02252-f001:**
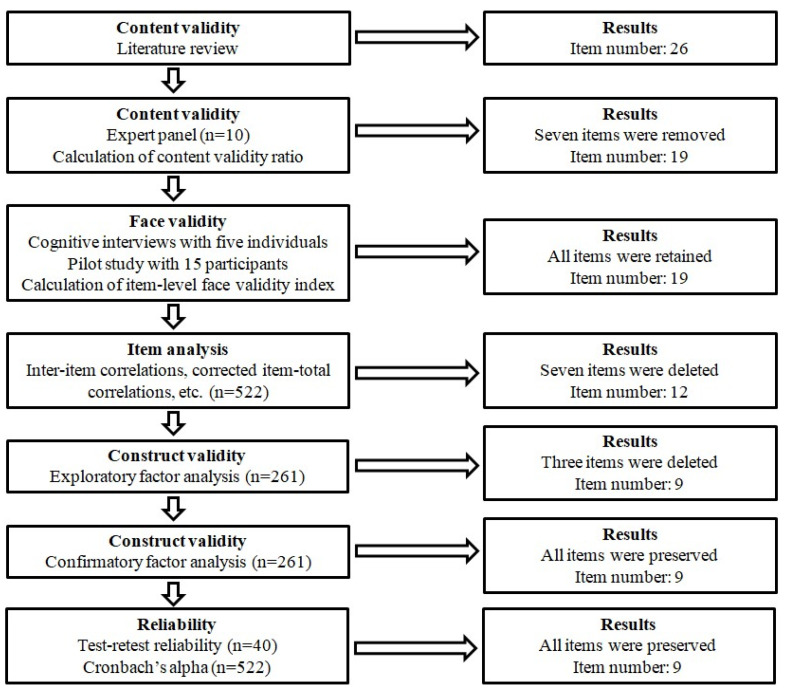
Development and validation of the Online Misinformation Susceptibility Scale.

**Figure 2 healthcare-13-02252-f002:**
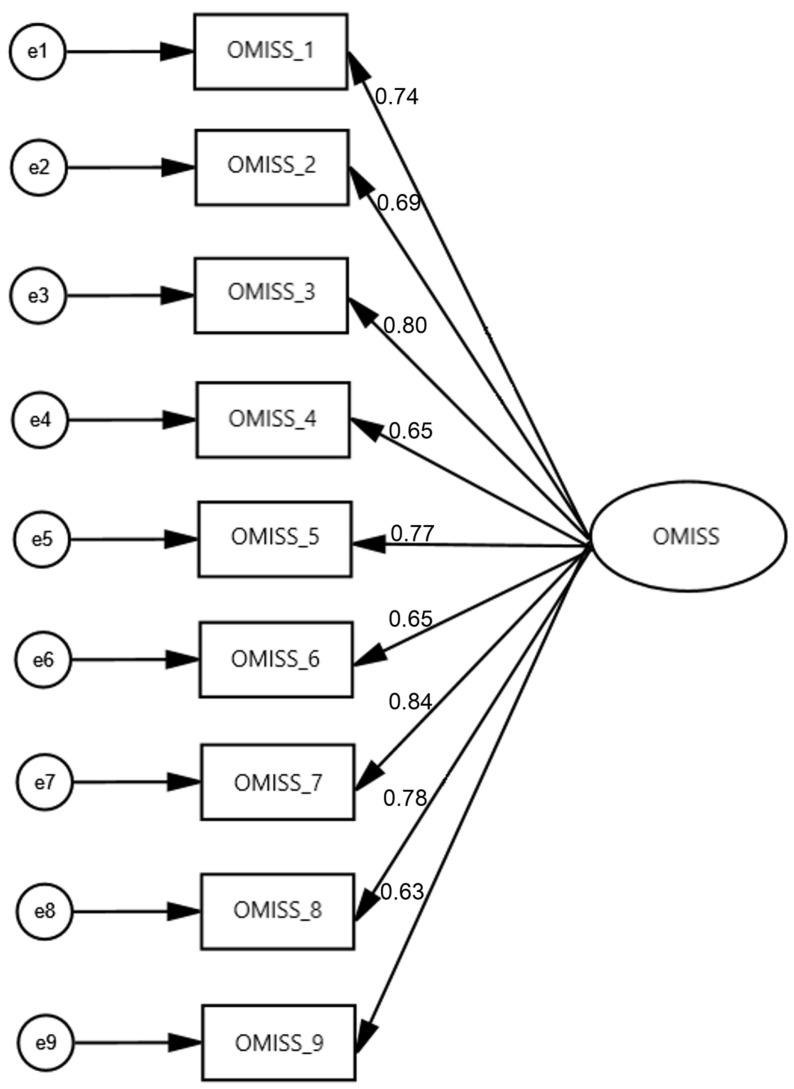
Confirmatory factor analysis of the Online Misinformation Susceptibility Scale.

**Table 1 healthcare-13-02252-t001:** Item analysis of the 19 items selected after the initial development phase of the Online Misinformation Susceptibility Scale (n = 522).

Please Think About What You Do When You See a Post or Story That Interests You on Social Media or Websites. How Often Do You …	Mean (Standard Deviation)	Corrected Item-Total Correlation	Floor Effect (%)	Ceiling Effect (%)	Skewness	Kurtosis	Cronbach’s Alpha if Item Deleted	Item Excluded or Retained
read the post in full?	2.64 (0.94)	0.430	12.6	1.0	−0.06	−0.62	0.885	Retained
2. share the post without read it in full?	1.25 (0.67)	0.245	85.1	0.6	3.01	9.36	0.889	Excluded
3. make a comment for the post without read it in full?	1.21 (0.58)	0.226	85.4	0.2	3.17	10.82	0.890	Excluded
4. check the website domain and URL?	3.19 (1.34)	0.665	14.9	19.7	−0.23	−1.12	0.877	Retained
5. check the publication date of the post?	2.69 (1.24)	0.664	21.5	8.2	0.19	−0.98	0.877	Retained
6. check if the post has been updated?	1.22 (0.64)	0.260	86.4	0.6	3.33	11.76	0.889	Excluded
7. check if the post includes reliable links and references such as scientific articles?	2.90 (1.25)	0.769	15.9	12.3	0.07	−0.99	0.873	Retained
8. check the post for grammatical, spelling, or expression errors?	2.69 (1.30)	0.640	25.3	9.8	0.17	−1.09	0.878	Retained
9. check if the post includes the author’s name?	2.93 (1.25)	0.728	15.7	11.7	0.01	−1.03	0.875	Retained
10. seek more information about the author of the post?	3.55 (1.14)	0.600	5.2	23.0	−0.48	−0.61	0.880	Retained
11. check if it is possible to contact the author of the post (e.g., if his/her email address is available)?	1.19 (0.56)	0.234	87.7	0.2	3.43	12.52	0.889	Excluded
12. check if the post solely expresses the opinion or experiences of the author?	1.14 (0.45)	0.234	89.5	0.0	3.50	12.75	0.889	Excluded
13. read the comments that the post received?	2.64 (1.15)	0.433	19.2	5.6	0.19	−0.79	0.886	Retained
14. check if the post originates from a reliable source, such as authoritative news sites?	2.62 (1.19)	0.790	20.9	6.7	0.27	−0.85	0.872	Retained
15. check if the post is reliable by searching other reliable sources on the web?	2.87 (1.17)	0.705	12.6	9.8	0.15	−0.81	0.876	Retained
16. check the website design?	3.51 (1.23)	0.590	7.1	25.9	−0.43	−0.83	0.880	Retained
17. discuss the post with someone you consider to be an expert?	3.41 (1.15)	0.504	5.0	20.5	−0.23	−0.83	0.883	Retained
18. use the Google image search to search if the post is true?	1.23 (0.61)	0.277	86.0	1.1	2.75	6.79	0.889	Excluded
19. check if the photos and videos in the post are real?	1.16 (0.50)	0.236	89.5	0.4	3.27	10.15	0.889	Excluded

**Table 2 healthcare-13-02252-t002:** Exploratory factor analysis using oblique rotation (Promax method) for the 12 items that were selected after the item analysis for the Online Misinformation Susceptibility Scale (n = 261).

Please Think About What You Do When You See a Post or Story That Interests You on Social Media or Websites. How Often Do You …	One Factor	
	Factor Loading	Communality
read the post in full?	0.513	0.264
2. check the website domain and URL?	0.770	0.593
3. check the publication date of the post?	0.721	0.520
4. check if the post includes reliable links and references such as scientific articles?	0.801	0.642
5. check the post for grammatical, spelling, or expression errors?	0.682	0.466
6. check if the post includes the author’s name?	0.781	0.611
7. seek more information about the author of the post?	0.692	0.479
8. read the comments that the post received?	0.545	0.297
9. check if the post originates from a reliable source, such as authoritative news sites?	0.849	0.721
10. check if the post is reliable by searching other reliable sources on the web?	0.799	0.638
11. check the website design?	0.693	0.480
12. discuss the post with someone you consider to be an expert?	0.588	0.345

**Table 3 healthcare-13-02252-t003:** Exploratory factor analysis using oblique rotation (Promax method) for the nine items that were selected after the first exploratory factor analysis for the Online Misinformation Susceptibility Scale (n = 261).

Please Think About What You Do When You See a Post or Story That Interests You on Social Media or Websites. How Often Do You …	One Factor	
	Factor Loading	Communality
check the website domain and URL?	0.771	0.595
2. check the publication date of the post?	0.778	0.606
3. check if the post includes reliable links and references such as scientific articles?	0.887	0.770
4. check the post for grammatical, spelling, or expression errors?	0.761	0.580
5. check if the post includes the author’s name?	0.841	0.707
6. seek more information about the author of the post?	0.748	0.559
7. check if the post originates from a reliable source, such as authoritative news sites?	0.884	0.781
8. check if the post is reliable by searching other reliable sources on the web?	0.824	0.679
9. check the website design?	0.724	0.524

**Table 4 healthcare-13-02252-t004:** Correlations and effect sizes between the Online Misinformation Susceptibility Scale, and the fake news detection scale, the Trust in Scientists Scale, the single-item trust in scientists scale, the Conspiracy Mentality Questionnaire, and the single-item conspiracy belief scale scores (n = 522).

Scale	Online Misinformation Susceptibility Scale		
	Pearson’s Correlation Coefficient	*p*-Value	Effect Size (%)
Fake news detection scale	−0.135	0.002	1.83
Trust in Scientists Scale	−0.304	<0.001	9.24
Single-item trust in scientists scale	−0.280	<0.001	7.84
Conspiracy Mentality Questionnaire	0.159	<0.001	2.53
Single-item conspiracy belief scale	0.095	0.030	0.91

**Table 5 healthcare-13-02252-t005:** Configural measurement invariance for the one-factor confirmatory factor analysis model with respect to gender, age, and daily usage time for social media/websites.

Model	RMSEA	GFI	NFI	CFI
Gender	0.045	0.955	0.956	0.984
Age	0.049	0.950	0.954	0.981
Daily usage time for social media/websites	0.051	0.947	0.952	0.980

## Data Availability

The original data presented in the study are openly available in FigShare at https://doi.org/10.6084/m9.figshare.29432735 (accessed on 25 June 2025).

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
