# Peer review of "The Online Misinformation Susceptibility Scale: Development and Initial Validation"

_healthcare, 2025, doi:10.3390/healthcare13172252_

Round 1

Reviewer 1 Report

Comments and Suggestions for Authors

The manuscript entitled “The Online Misinformation Susceptibility Scale: Development and initial validation” was interesting. The authors developed and validated a tool to measure susceptibility to online misinformation, i.e. the Online Misinformation Susceptibility Scale (OMISS). However, the following issues need further attentions.

  • In the methods section, Research design should be explained.
  • “Thorough literature review” is not clear, what type of literature review has been conducted? What was the methodology?
  • What was the result for calculating Cronbach’s alpha? Please add it to Figure 1.
  • The final scale has limited number of questions and need to be revised.
  • The discussion section is a bit long, and include repetition of the research procedures. This section needs to be revised.
  • Although the researchers developed a new scale, I think the paper is more suitable to be submitted to other relevant journals.

Additional comments

Overall, in my opinion, the title “The Online Misinformation Susceptibility Scale: Development and initial validation” is general and as the authors noted “the scale is a general-context measure that can measure online misinformation susceptibility in all fields”. It has not been specifically developed for the healthcare domain. The developed scale can be used for any online misinformation susceptibility, and the manuscript can be sent to the journals which are more relevant to information and communication. In addition, it is not clear what type of literature review has been conducted and how the scale items have been initially selected. The final scale has a few questions which suggest that the evaluation results may not be strong enough for further analysis and comparisons. Moreover, if we identify the level of susceptibility, how can we help people? Can we stop getting access to misinformation? Or can we stop dissemination of misinformation? So, the research implications are not clear in this manuscript.

Reviewer 2 Report

Comments and Suggestions for Authors

The paper is methodologically rigorous, theoretically grounded, and offers a validated tool of practical significance. Minor edits to improve clarity, depth in the discussion of limitations, and enhancement of statistical reporting would strengthen the final version.

  • The abstract would benefit from a more concise statement on the theoretical novelty of the scale
  • Consider structuring the introduction more clearly around the “problem–gap–hook–solution” format to aid readability.
  • The recruitment process through social media may introduce sampling bias. Consider discussing limitations in greater depth.
  • Clarify if attention checks or data quality screening were used.
  • Consider reporting measurement invariance across demographic groups
  • Include effect sizes in the correlation table (Table 4) to better interpret the strength of associations.
  • Expand on limitations of self-report measures
  • Recommend longitudinal validation or use of digital behaviour data to triangulate self-reports.
  • Some sentences are dense. Consider breaking complex ideas into two sentences for better readability.
  • Remove duplicate or excessive references (e.g., refs [32–37] appear multiple times).
  • Tighten language in the Discussion to reduce redundancy.
  • Replace phrases like “our study shows” with “this study shows” for a more objective tone.

Reviewer 3 Report

Comments and Suggestions for Authors

Overall

The article addresses a relevant topic that aligns well with the scope and objectives of the journal. The idea behind the manuscript is clearly articulated, interesting, and of current scientific interest. 

The Title

The title expresses clearly what the manuscript is about. It contains the necessary keywords to make it discoverable by a reader in this field.

Abstract

The abstract summarizes the objectives, research methods, novel contribution, and conclusions. But it is too long. It should be shortened and limited to 250 words.

The introduction provides a concise overview of the current state of research on the topic and establishes the context for the study

Methods

The research design and methods are appropriate for the research question.

 It is clear how the research data were collected.  The study methods are explained clearly.

Results.

The results are presented with accuracy, and the calculations appear to be correct.

The conclusions are accurate and supported by the content.

Additional comments

1. The abstract is too long and needs to be rewritten so that it does not exceed 250 words.

2. The introduction should clearly state the research question and the initial hypothesis.

3. The uniqueness of the study should also be mentioned in the conclusions.

Reviewer 4 Report

Comments and Suggestions for Authors

The paper presents a significant contribution to the field of measuring susceptibility to online misinformation, a topic of great relevance and social relevance. The decision to develop a short, clear, and well-validated scale like the OMISS addresses a specific need in research and practical interventions, offering a reliable and easily usable tool in multidisciplinary contexts.

Particularly noteworthy are the thoroughness of the validation process, which includes both exploratory and confirmatory factor analyses, assessments of concurrent validity and test-retest reliability, and the multidisciplinary approach adopted from the initial item development stage. This lends the work methodological soundness and scientific rigor.

An aspect that could further strengthen the manuscript concerns the discussion of the scale's potential limitations, particularly regarding its applicability in different cultural and linguistic contexts. It would be useful to explore how the OMISS could be adapted or validated in populations with diverse sociocultural backgrounds to broaden its scope and generalizability. Furthermore, a more detailed discussion of how the scale can be integrated into practical digital literacy interventions and concrete examples of its application could increase the impact and practical utility of the work.

In conclusion, the manuscript is well-structured, with clear and convincing results, and represents an important step forward for research and practice in countering online disinformation. I therefore recommend publication, subject to the considerations outlined above.

Round 2

Reviewer 1 Report

Comments and Suggestions for Authors

I appreciate the authors for their time and efforts to revise the manuscript.  However, one minor issue is remaining. If the authors conducted a systematic review and published its results, it should be referenced. If they have not published the results of the systematic review, more details about the systematic review methodology (e.g, search strategies, databases, critical evaluation tools, risk of bias and quality assessment, etc.) and results of this review should be added to the manuscript.

Author Response

Comment

I appreciate the authors for their time and efforts to revise the manuscript.  However, one minor issue is remaining. If the authors conducted a systematic review and published its results, it should be referenced. If they have not published the results of the systematic review, more details about the systematic review methodology (e.g, search strategies, databases, critical evaluation tools, risk of bias and quality assessment, etc.) and results of this review should be added to the manuscript.

Response: Done

Dear Reviewer, thank you especially for this comment.

We have not published the results of the systematic review since our aim was to identify potential items to include in our scale. Thank you for your comment to add more details about the systematic review methodology. We agree with your comment. It was our oversight that we did not mention these details.

Thus, we add the following text in the section 2.1. Development of the scale.

…In particular, we searched on Pubmed and Scopus using the following strategy in title: (((misinformation) OR ("fake news")) OR (disinformation)) OR (malinformation). We employed the Preferred Reporting Items for Systematic Reviews and Meta-Analysis (PRISMA) guidelines to conduct our review [53]. We examined studies that explore the issues of misinformation, fake news, disinformation and malinformation. Moreover, we examined articles published in English and in journals with a peer review system. We searched Pubmed and Scopus from their inception until May 10, 2025. We identified 2686 articles through Pubmed and 13,620 articles through Scopus. Emphasis was given to articles that measure misinformation, fake news, disinformation or malinformation. Three independent authors performed the review. First, they removed duplicates and then examined titles and abstracts to identify relevant articles. Our inclusion criteria were the following: (a) articles that examine misinformation, fake news, disinformation or malinformation, (b) articles in English, and (c) articles in journals with peer review system. Three independent authors extracted data from studies, such as authors, year of publication, country, study design, study population, sampling method, data collection time, and results. The Joanna Briggs Institute critical appraisal tools [54] were employed to examine the quality of the studies. These risk of bias tools examine several aspects of studies quality, such as study settings, inclusion criteria, measurement of exposure and outcome, measurement of confounding, and statistical analysis. After all, quality of studies is classified as poor, moderate or high…

Dear Reviewer, we are grateful for your valuable input and guidance. We sincerely appreciate your support.

Warm regards,

The Authors